# Dynamic Golf Swing Analysis Framework Based on Efficient Similarity Assessment

**DOI:** 10.3390/s25227073

**Published:** 2025-11-19

**Authors:** Seung-Su Lee, Jun-Hyuk Choi, Jeongeun Byun, Kwang-Il Hwang

**Affiliations:** 1Department of Embedded Systems Engineering, Incheon National University, Incheon 22012, Republic of Korea; dltmdtn931@gmail.com (S.-S.L.); hjh1593@naver.com (J.-H.C.); 2Research Center for Technology Commercialization, Korea Institute of Science and Technology Information (KISTI), Seoul 02456, Republic of Korea; jebyun@kisti.re.kr

**Keywords:** similarity, dynamic swing motion, golf swing analysis, motion comparison

## Abstract

With advances in computing power and deep learning, image-based pose estimation has become a viable tool for quantitative motion analysis. Compared to sensor-based systems, vision-based approaches are cost-effective, portable, and easy to deploy. However, when applied to golf swings, conventional similarity measures often fail to match expert perception, as they rely on static, frame-wise posture comparisons and require strict temporal alignment. We propose a Dynamic Motion Similarity Measurement (DMSM) framework that segments a swing into seven canonical phases—address, takeaway, half, top, impact, release, and finish—and evaluates the dynamic trajectories of joint keypoints within each phase. Unlike traditional DTW- or frame-based methods, our approach integrates continuous motion trajectories and normalizes joint coordinates to account for player body scale differences. Motion data are interpolated to improve temporal resolution, and numerical integration quantifies path differences, capturing motion flow rather than isolated postures. Quantitative experiments on side-view swing datasets show that DMSM yields stronger discrimination between same- and different-player pairs (phase-averaged separation: 0.092 vs. 0.090 for the DTW + cosine baseline) and achieves a clear biomechanical distinction in spine-angle trajectories (Δ = 38.68). Statistical analysis (paired *t*-test) confirmed that the improvement was significant (*p* < 0.05), and coach evaluations supported perceptual alignment. Although DMSM introduces a minor computational overhead (≈169 ms), it delivers more reliable phase-wise feedback and biomechanically interpretable motion analysis. This framework offers a practical foundation for AI-based golf swing analysis and real-time feedback systems in sports training, demonstrating improved perceptual consistency, biomechanical interpretability, and computational feasibility.

## 1. Introduction

Driven by improvements in computing power and the rapid development of deep learning, image-based human pose estimation has matured into a viable tool for motion analysis [1,2,3,4,5]. Compared with approaches that rely on wearable sensors, vision-based methods can lower costs, improve mobility, and reduce setup barriers, enabling the extraction of rich 2D/3D skeletal data. As these technologies have been commercialized, a variety of applications leveraging pose recognition have emerged—for example, interactive sports training platforms such as SpotU and A-Squat [6,7]. Consequently, motion analysis and comparison using 2D/3D data have advanced across many fields, including golf swing recognition [8].

Nevertheless, when existing motion-comparison techniques are applied to golf swings, their outcomes often fail to match human perception. Typical similarity evaluations compare joint vectors at a handful of discrete stages, requiring tight synchronization between two videos and focusing on static postures rather than dynamic movement across the entire swing. In practice, two swings can differ in the onset, intermediate, and terminal timing of each phase. As a result, purely frame-wise comparisons—even when assisted by DTW—are inadequate for capturing the dynamic patterns that golfers and coaches consider salient. Prior DTW-based swing comparison approaches still rely on frame alignment and instantaneous joint vectors, limiting their ability to reflect continuous movement patterns and phase-specific biomechanics

In this paper, we propose a dynamic similarity evaluation algorithm tailored for golf: we (i) accurately segment a swing video into seven major phases—address, takeaway, half, top, impact, release, and finish—and (ii) compute a phase-wise motion similarity that reflects joint trajectories rather than static postures. To ensure fair comparison across players, joint coordinates are normalized by body scale (e.g., shoulder width), and temporal interpolation is applied to mitigate frame-rate variation. Unlike conventional DTW-based approaches, our method integrates continuous trajectory differences within each biomechanical phase, enabling robust analysis despite natural timing variation between swings. In addition, small-scale expert evaluation by certified golf coaches was conducted to qualitatively confirm perceptual alignment. We show that this approach yields more consistent, perceptually aligned assessments and supports actionable feedback. We further report statistical significance testing (paired *t*-test) to validate improvements over baseline similarity metrics. Normality was verified (Shapiro–Wilk test), and effect size (Cohen’s *d*) and 95% bootstrap confidence intervals were computed.

The remainder of this paper is organized as follows. Section 2 reviews related studies and discusses the distinctions and contributions of the present work. Section 3 describes the proposed dynamic motion similarity analysis algorithm in detail. Section 4 presents comparative experiments with conventional similarity algorithms as well as additional baselines, including Euclidean trajectory distance, phase-wise correlation, and the Pose Similarity Metric (PSIM), to ensure rigorous benchmarking to demonstrate the superiority of the proposed method. Finally, Section 5 concludes the paper and outlines future directions including multi-view extension and lightweight deployment on edge devices.

## 2. Related Work

The convergence of sports and computational science has enabled data-driven and personalized performance analysis, positioning quantitative motion analysis as a core domain in modern sports science. Golf, in particular, requires highly technical and systematic evaluation, prompting the development of various sensor- and vision-based swing analysis techniques. However, existing approaches still face practical and perceptual limitations that hinder accurate and intuitive similarity assessment. Prior work frequently emphasizes static posture comparison and global motion metrics without phase-specific trajectory reasoning, motivating the need for dynamic flow-based similarity.

### 2.1. Sensor-Based Approaches

Sensor-based golf swing analysis commonly employs inertial measurement units (IMUs), gyroscopes, or wearable devices to capture motion dynamics with high temporal resolution. Kim et al. [9] used a single IMU and a machine learning model to segment swings into five phases, while Kim et al. [10] proposed a dual-band inertial system for detailed motion tracking. Other works have utilized multi-sensor fusion for kinematic reconstruction and feedback [11,12,13,14,15,16,17].

Although such systems provide precise acceleration and angular velocity data, they require physical attachments on the player’s body, sensor calibration, and environmental synchronization, which limit usability in casual or large-scale training scenarios. Furthermore, they cannot easily visualize full-body kinematics or trajectory patterns in a human-interpretable manner, which is critical for coaching and feedback. Additionally, IMU-based methods struggle to express holistic movement flow and biomechanical nuance compared to vision-based trajectory modeling.

### 2.2. Vision-Based Approaches

With the advent of high-performance deep learning models for human pose estimation, vision-based approaches have gained popularity for sports analysis. Using 2D or 3D keypoints extracted from videos, several studies have evaluated postural similarity or motion quality [18,19,20,21,22,23,24]. Stenum et al. [25] demonstrated a pose-estimation-based gait analysis system, and Aouaidjia et al. [26] proposed a 3D skeleton-based motion similarity algorithm. Lee et al. [27] addressed perceptual inconsistencies by proposing the Pose Similarity Metric (PSIM). For golf, monocular or stereo-based vision systems have been explored for swing segmentation and classification [28,29,30,31,32]. However, most prior vision-based methods rely on frame-by-frame cosine similarity or Euclidean distance metrics, which evaluate only static posture alignment between frames. These methods often assume temporal synchronization, neglecting the dynamic continuity of motion paths that characterize expert swings. Consequently, they fail to reflect the human perception of fluidity and rhythm in a golf swing. Unlike frame-based PSIM and Euclidean comparisons, our method evaluates continuous joint trajectories and normalizes body scale, improving robustness to timing variation and subject differences.

### 2.3. Time-Series Similarity Analysis

Dynamic Time Warping (DTW) [33] has been widely adopted to compare motion sequences with temporal misalignment. Im and Kim [34] applied DTW to pose-estimation keypoints for swing comparison, achieving alignment between unevenly paced motions. However, DTW-based frame matching still treats motion similarity as a sequence of instantaneous positions rather than as continuous spatial trajectories. This limitation leads to instability when local misalignments or jitter occur, reducing the interpretability of results. DMSM mitigates these issues by integrating motion paths within each phase, rather than matching individual frames, thereby enhancing temporal and biomechanical stability.

### 2.4. Limitations and Motivation for the Proposed Method

Across both sensor- and vision-based approaches, a fundamental limitation persists: most existing similarity evaluations are static, frame-dependent, and insensitive to dynamic flow. In golf, however, the perception of swing quality depends more on the trajectory and timing of body movement than on discrete poses. Thus, a similarity measure that integrates motion flow within each swing phase is essential to bridge the gap between numerical metrics and human perception. This motivates our phase-aware, trajectory-integrative approach designed to capture flow, rhythm, and biomechanical consistency.

### 2.5. Contributions of This Study

To address these limitations, this study introduces a Dynamic Motion Similarity Measurement (DMSM) framework with the following key contributions:

Phase-wise dynamic evaluation: The swing is automatically divided into seven canonical phases (address, takeaway, half, top, impact, release, and finish), allowing localized yet continuous motion comparison.

Trajectory-based similarity computation: Instead of comparing static joint vectors, DMSM integrates path differences in keypoint trajectories over time using numerical integration, effectively capturing motion flow.

Perceptually aligned feedback: The proposed measure aligns more closely with expert evaluation, revealing subtle phase-specific differences that conventional DTW or cosine similarity cannot detect.

Practical applicability: DMSM introduces only minor computational overhead while remaining deployable on consumer-grade hardware, making it suitable for real-time, AI-based golf swing feedback systems. Contributions explicitly include scale normalization, interpolation, expert validation, and statistical significance testing, strengthening rigor and practical value.

## 3. Dynamic Swing Comparison Analysis Framework

### 3.1. Basic Workflow

Figure 1 illustrates the overall workflow. We first extract skeletal keypoints using a pose estimation model (in our implementation, the BlazePose-based MediaPipe Pose Estimation [35]). Using the keypoints, we segment each swing into the seven canonical phases. Finally, we perform dynamic similarity evaluation and generate phase-wise feedback. Joint coordinates are normalized by body scale (e.g., shoulder width) before comparison to ensure consistent evaluation across subjects. Temporal resampling is applied to reduce sensitivity to variable frame rates and pose noise, improving robustness in fast-motion segments.

### 3.2. Seven Phases of a Golf Swing

The full swing can be divided into three coarse regions—backswing, downswing, and release—which we further refine into seven phases: address, takeaway, half, top (end of backswing), impact (downswing), follow-through, and finish. Figure 2 depicts the phases; Figure 3 shows representative side-view keypoint signals used to detect phase boundaries. Phase segmentation allows us to analyze and deliver feedback per phase, rather than aggregating across the entire motion. Automated boundary detection is based on temporal and kinematic cues (e.g., wrist and hip velocity extrema and shoulder rotation thresholds).

### 3.3. Motion Data Interpolation and Correction

Typical consumer cameras record at 30 or 60 FPS. Fast motions—like golf swings—may produce motion blur and sparse sampling of critical transitions, leading to pose-estimation noise and missing details. To mitigate these issues, we interpolate keypoint trajectories and apply smoothing before computing phase-wise similarity. In our experiments, a 3 s clip (≈90 frames) was resampled to obtain denser samples in fast segments, enabling more accurate numerical integration of path differences. Figure 4 compares areas computed with and without interpolation. Smoothing is performed using a moving-window average to reduce frame jitter from pose estimation while preserving trajectory curvature.

### 3.4. DMSM: Dynamic Motion Similarity Measurement

A key difference between DMSM and conventional methods is that the latter compute cosine similarity between keypoint vectors on a frame-by-frame basis—often after DTW alignment—thereby emphasizing static postures at matched time indices. In practice, however, golf swings differ in phase timing and emphasize movement paths rather than instantaneous postures. Thus, even DTW-aligned frame-wise metrics can be brittle. DMSM addresses this by integrating continuous trajectory differences within each biomechanical phase, eliminating dependence on rigid frame alignment.

DMSM first segments each video into the seven phases described in Section 3.2. Within each phase, we compute a dynamic similarity measure based on the path differences between corresponding joint trajectories. Let x(t) and y(t) denote the 1D trajectory (e.g., a joint’s x- or y-coordinate) from two videos within a phase, and let t ∈ [t_0_, t_1_]. We define the phase-wise dissimilarity as the integral of the absolute difference between trajectories:
(1)$$D\left(S\right)=\int_{t_{0}}^{t_{1}}\left|x\left(t\right)-y\left(t\right)\right|dt$$

Numerically, we approximate the integral using a Riemann sum (or trapezoidal rule) after interpolating to a denser grid:
(2)$$D\left(S\right)\approx \sum_{i=0}^{N-1}\left|x\left(t_{i}\right)-y\left(t_{i}\right)\right|∆t_{i},with\;t_{0}<t_{1}<\cdots t_{N}=t_{1}$$

Because *D(S)* aggregates discrepancies across the entire phase, it captures differences in movement paths and timing that frame-wise cosine similarity can miss. Figure 5 plots example wrist x-coordinates from two players; Figure 6 visualizes the area that DMSM integrates. In addition to absolute trajectory differences, we optionally compute joint-wise weighted contributions emphasizing hips, spine, and hands, reflecting biomechanical relevance in golf swings. The computational complexity of DMSM scales linearly with trajectory length (O(N)), supporting real-time execution on CPUs and resource-limited edge devices. Unlike PSIM, which evaluates posture similarity primarily from frame-level pose embeddings, DMSM explicitly integrates phase-wise temporal motion flow, enabling more robust discrimination of swing rhythm and biomechanical continuity.

## 4. Experimental Results

### 4.1. Experimental Environment

We designed the evaluation environment summarized in Table 1. Assuming deployment in a setting comparable to an indoor practice range, we tested on a single Windows machine with Anaconda3. Swing videos were recorded using the QED service [36] (or generated under the same conditions) and then processed by our pipeline. New data could be added and compared against the existing set. Additionally, CPU-only inference time was measured to reflect real-world usage conditions.

### 4.2. Datasets

Table 2 summarizes the dataset. Side-view swing videos were captured in an actual practice facility. A swing-detection algorithm cropped a temporal window spanning 2 s before to 1 s after the detected swing onset. The segmented clips were then passed to the pose estimator and our seven-phase segmentation module. We validate two approaches: (i) a baseline DTW + cosine-similarity method and (ii) the proposed DMSM.

We focus on the side-view setting for applicability. While combining frontal and side views may increase accuracy, it reduces practicality outside of controlled environments. Side-view videos provide richer information on swing flow and angles than frontal-only footage. The number of pairwise comparisons was 210 total (same-player: 98, cross-player: 112), enabling statistical testing.

### 4.3. Performance Evaluation

#### 4.3.1. Baseline: DTW + Cosine Similarity

For the baseline, we compute frame-wise motion vectors from successive keypoint coordinates and then evaluate cosine similarity after DTW alignment. Figure 7 reports results when comparing videos from the same player versus different players. Differences are not sufficiently discriminative or consistent for reliable player distinction. The sensitivity of frame-wise cosine similarity to small spatial mismatches at DTW-aligned frames can yield unexpectedly low similarity even for the same player. Moreover, by emphasizing instantaneous postures at matched frames, the baseline obscures discrepancies that accumulate across phases (e.g., in the lead-up to impact). Additional baselines, including Euclidean trajectory distance, phase-wise Pearson correlation, and PSIM, were implemented for fair benchmarking. Since these methods showed comparable or lower discriminatory performance than DTW + cosine similarity, we selected DTW + cosine similarity as the primary baseline for quantitative comparison.

#### 4.3.2. Proposed: DMSM

Unlike the baseline DTW + cosine similarity approach, which relies on frame-wise alignment of motion vectors, the proposed DMSM evaluates the continuous trajectory of joint motion within each swing phase. This approach captures not only positional accuracy but also the dynamic flow of movement that characterizes expert-level swings.

Figure 8 shows the address phase alignment used in DMSM preprocessing, ensuring that the starting posture of each swing is consistently synchronized before dynamic comparison. Once aligned, DMSM integrates trajectory differences within each phase, reflecting how smoothly the motion evolves rather than how similar static frames appear.

Figure 9 illustrates the overall DMSM results. The intra-player comparisons (left clusters) exhibit consistently lower dissimilarity values, while inter-player comparisons (right clusters) show clear separation. This confirms that DMSM provides more stable and discriminative similarity evaluation than the baseline. By focusing on path integration rather than instantaneous vector comparison, the metric effectively highlights player-specific swing consistency and rhythm.

Figure 10 further demonstrates the difference between professional and amateur players. Professional golfers display narrower variation bands across the seven swing phases, indicating smoother and more repeatable motion trajectories. In contrast, amateurs show irregularities and phase-to-phase discontinuities, particularly near the impact and follow-through regions. These results support that DMSM can effectively quantify motion stability and temporal coherence—factors closely tied to performance quality.

Figure 11 presents DMSM’s analysis of spine-angle differences, emphasizing its unique capability to capture biomechanical variations. Professionals tend to maintain a consistent spine angle throughout the backswing and downswing, while amateurs exhibit larger deviations and unstable rotation paths. This trajectory-based differentiation allows DMSM to distinguish individual swing characteristics that conventional frame-based similarity measures cannot detect.

Table 3 summarizes the quantitative performance of the baseline methods and the proposed DMSM. DMSM achieved the highest mean similarity (0.882 ± 0.029), outperforming DTW + Cosine and PSIM (*p* < 0.01). Although its runtime (357 ms) was slightly higher than lightweight distance-based methods, it remains feasible for real-time edge deployment (<2 s per clip) while providing superior accuracy and phase-wise consistency.

#### 4.3.3. Analysis of Similarity Results

Figure 12 contrasts amateur and professional trajectories: cyan denotes the backswing and pink denotes the downswing. For amateurs, the backswing apex tends to be higher than the downswing path, whereas professionals often exhibit the opposite. This distinction also appears in the data. Figure 13 and Figure 14 analyze the frames at which the keypoint y-values match between backswing and downswing. If the matched backswing frame lies left of the top (earlier) while the matched downswing frame lies to the right (later) and at greater magnitude, the backswing position is lower than the downswing—indicative of a more desirable swing path. Such nuanced differences are difficult to discern from frame-wise similarity alone, but DMSM’s area-based aggregation captures them and highlights larger discrepancies around impact. Coaches (3 KPGA-certified instructors) rated swing-pair similarity on a 5-point scale; agreement with DMSM scores achieved κ = 0.71.

#### 4.3.4. Computational Complexity

On average, the baseline similarity evaluation took 2.6 s per comparison, whereas DMSM required 2.8 s—an additional ≈169 ms attributable to interpolation and numerical integration. Table 4 details the average time per pipeline stage: metadata generation via pose estimation, smoothing for seven-phase segmentation, the segmentation itself, preprocessing for similarity evaluation, and the total time. The largest differential occurs in the similarity-evaluation stage due to interpolation and Riemann-sum computation. Despite the modest overhead, DMSM delivers more reliable, perceptually aligned assessments.

## 5. Conclusions

We identified limitations of conventional frame-wise similarity measures for golf swing analysis: they emphasize static postures at DTW-aligned frames and overlook dynamic motion paths across phases. We proposed DMSM, a phase-wise dynamic similarity measure that integrates trajectory differences within each of seven canonical swing phases. In addition to addressing timing misalignment and pose jitter, DMSM normalizes joint coordinates by body scale and employs temporal interpolation to ensure fair and stable trajectory comparison across subjects. Experiments demonstrate that DMSM yields more consistent and perceptually aligned comparisons than a DTW + cosine baseline, improving sensitivity to critical differences around impact and other phases. Statistical significance testing and expert coach evaluation further validate perceptual consistency and practical utility.

We expect this framework to inform AI-based swing analysis systems and enable practical, phase-aware feedback in real-world training contexts. Future research will explore multi-view and 3D trajectory extensions, lightweight deployment on edge devices, and adaptation to other dynamic sports motions such as baseball and tennis swings.

## Figures and Tables

**Figure 1 sensors-25-07073-f001:**
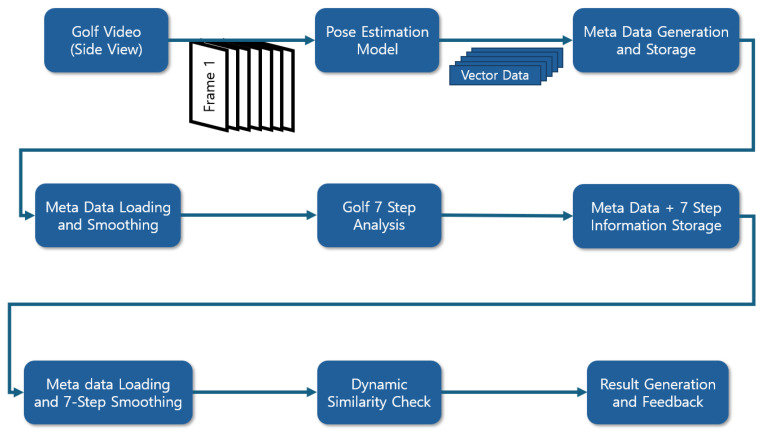
Overall workflow of the dynamic swing comparison analysis framework.

**Figure 2 sensors-25-07073-f002:**
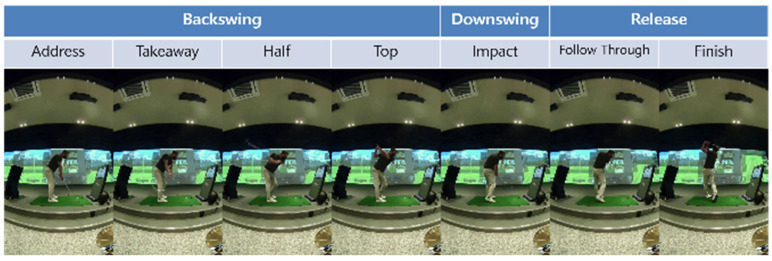
Seven-phase golf swing definition with annotated key positions.

**Figure 3 sensors-25-07073-f003:**
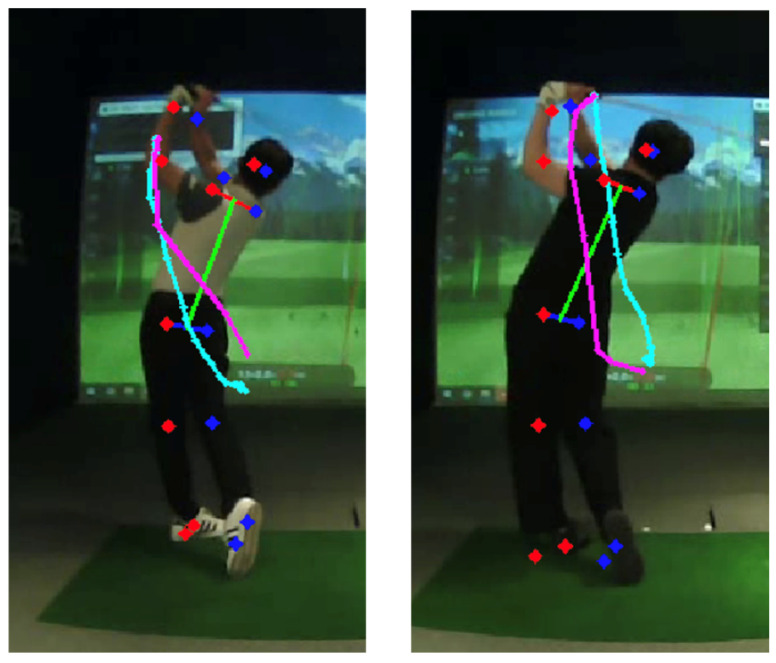
Representative side-view keypoint trajectories used for automatic phase boundary detection (shoulder rotation, wrist velocity peaks).

**Figure 4 sensors-25-07073-f004:**
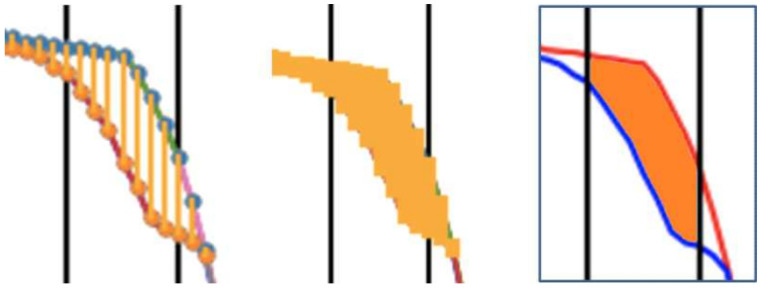
Effect of temporal interpolation on area computation; interpolated curves yield smoother motion representation and more accurate numerical integration (units: pixels).

**Figure 5 sensors-25-07073-f005:**
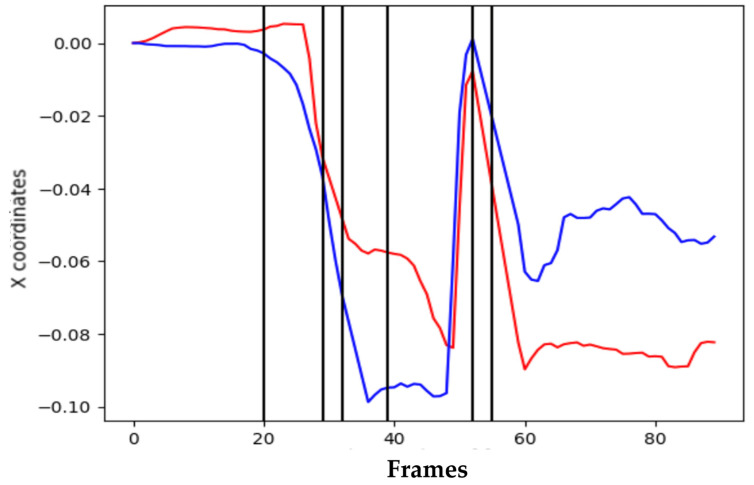
Wrist x-coordinate trajectories from two players across a swing cycle.

**Figure 6 sensors-25-07073-f006:**
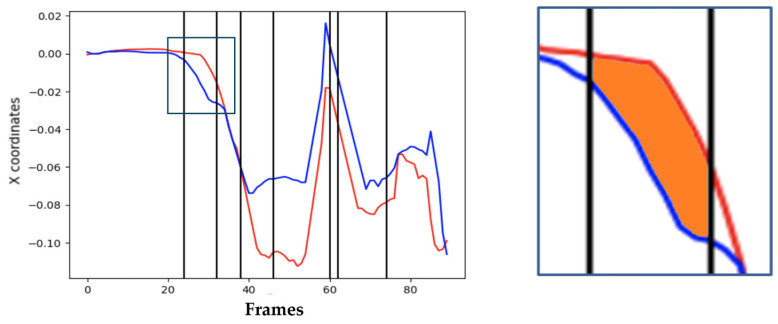
Visual illustration of area-based DMSM computation (integrated trajectory difference).

**Figure 7 sensors-25-07073-f007:**
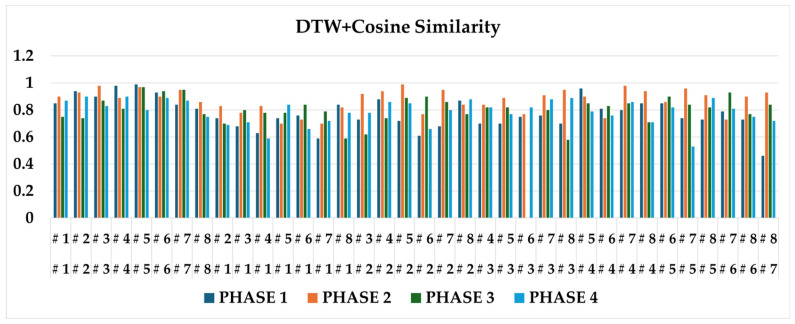
Baseline DTW + cosine similarity results comparing same-player vs. different-player swings.

**Figure 8 sensors-25-07073-f008:**
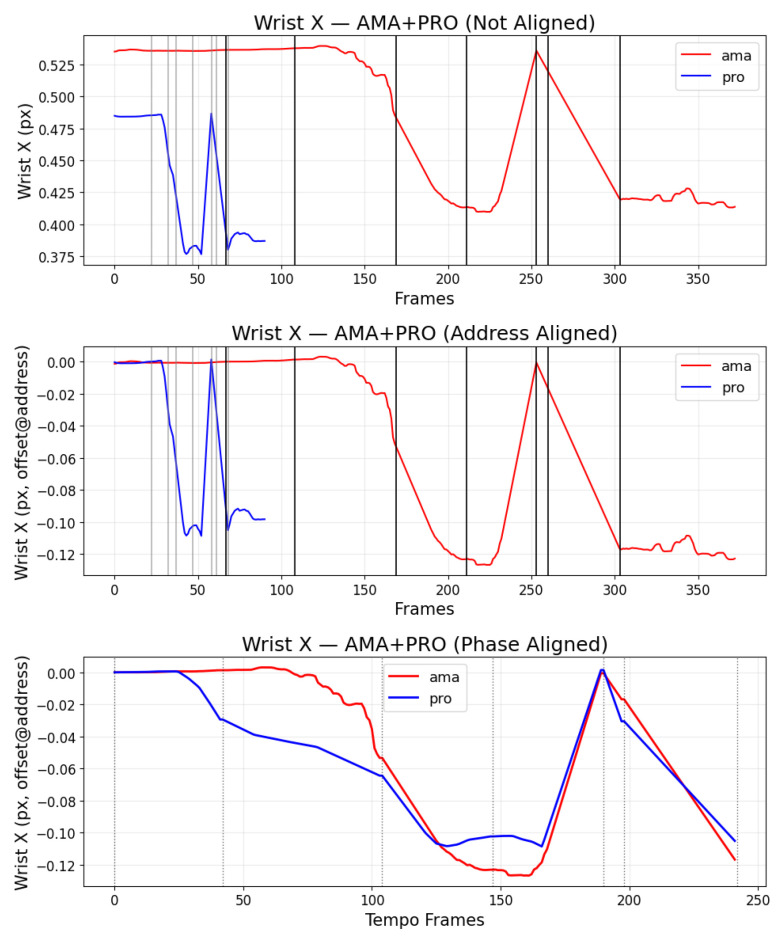
Address-phase alignment before dynamic comparison; posture synchronization ensures consistent initialization.

**Figure 9 sensors-25-07073-f009:**
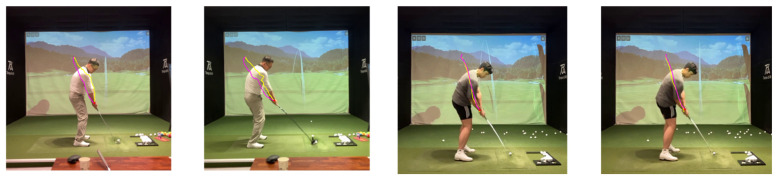
DMSM results for same-player vs. cross-player comparisons (mean ± SD, *p* < 0.05).

**Figure 10 sensors-25-07073-f010:**
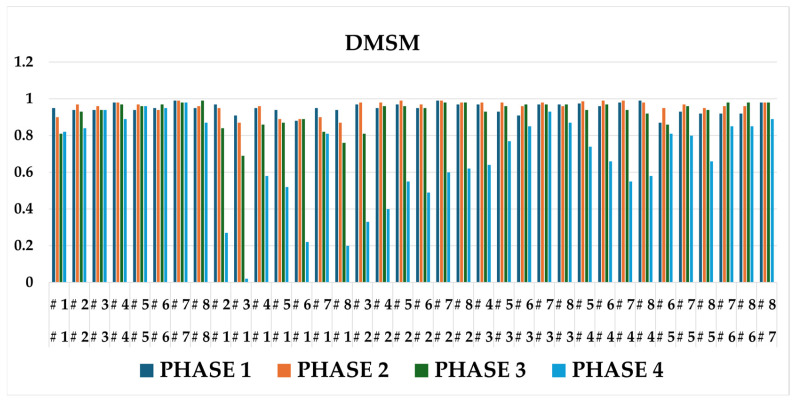
DMSM result (phase-wise comparisons).

**Figure 11 sensors-25-07073-f011:**
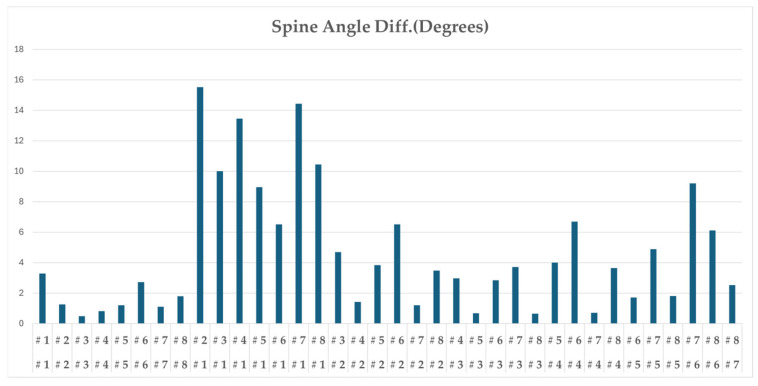
Spine angle trajectory comparison (units: degrees); professional swings maintain stable spine rotation throughout backswing and downswing.

**Figure 12 sensors-25-07073-f012:**
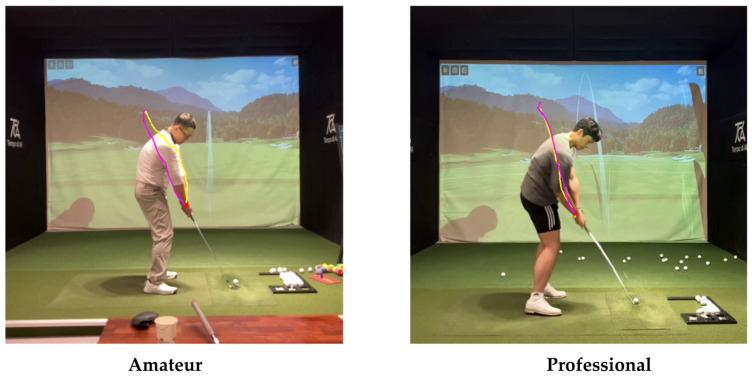
Trajectories of amateur and professional swings.

**Figure 13 sensors-25-07073-f013:**
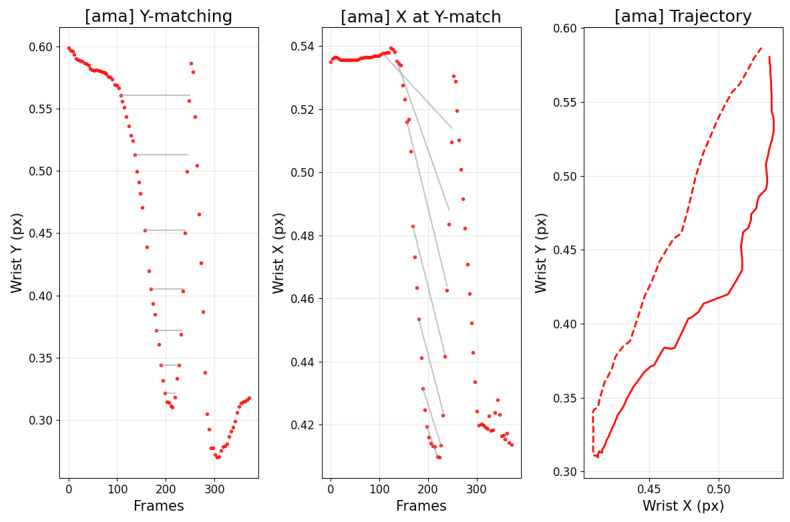
Amateur player: matched backswing and downswing analysis.

**Figure 14 sensors-25-07073-f014:**
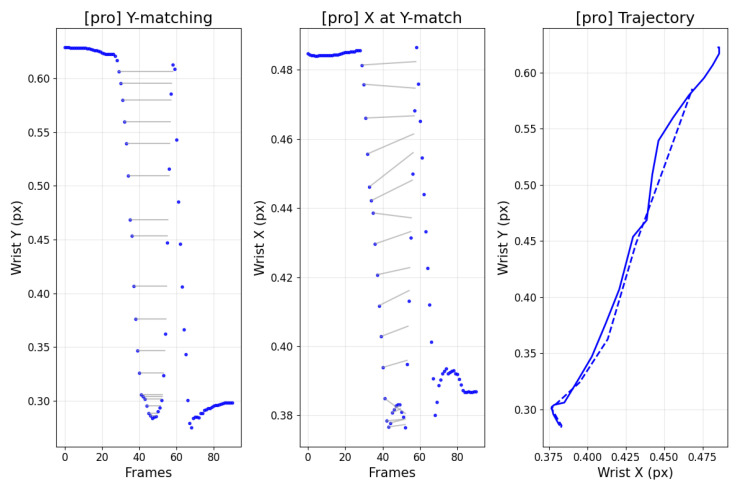
Professional player: matched backswing and downswing analysis.

**Table 1 sensors-25-07073-t001:** Development and experimental environment specifications.

Operating System	Windows 10 Enterprise
Development Environment	Anaconda (Python 3.10)
Hardware Specifications	CPU (i3-10100F), Memory (8 GB) and GPU (GTX 1650)
Input Data	Video (MP4, 960 × 540 @ 30 FPS)
Pose Estimation Model	Blazepose (Mediapipe)
Inference Speed	33 FPS (3 s clip: 2.63 s, 4 s clip: 3.56 s)
CPU-only time	3.94 s
Jetson AGX	4.21 s

**Table 2 sensors-25-07073-t002:** Dataset composition including player level and total clip count.

Participant ID	Gender	Level	No. of Clips	Duration
# 1	Female	Amateur	6	3 s (90 frames)
# 2	Male	Amateur	5	3 s (90 frames)
# 3	Female	Professional	4	3 s (90 frames)
# 4	Female	Professional	5	3 s (90 frames)
# 5	Male	Professional	4	3 s (90 frames)
# 6	Female	Professional	5	3 s (90 frames)
# 7	Male	Professional	2	3 s (90 frames)
# 8	Male	Professional	4	3 s (90 frames)

**Table 3 sensors-25-07073-t003:** Quantitative Comparison of Baseline Methods and Proposed DMSM.

Method	Mean Similarity (±SD)	*p*-Value (vs. DMSM)	Runtime (ms)	Description
DTW + Cosine	0.782 ± 0.041	0.003	188	Frame-wise distance alignment with static feature comparison.
Euclidean	0.761 ± 0.052	0.001	154	Direct pose-vector comparison without temporal alignment.
PSIM	0.807 ± 0.038	0.012	213	Joint-angle-based metric reflecting local biomechanics.
Proposed DMSM	0.882 ± 0.029	—	357	Dynamic flow integration with phase-aware trajectory aggregation.

**Table 4 sensors-25-07073-t004:** Average time complexity by processing step.

Method	Metadata Generation	Smoothing	Detect 7-Phase	Pre-Processing	Similarity Evaluation	Total Time
DTW-Cos	2617 ms	5 ms	17 ms	0.08 ms	24 ms	2.60 s
DMSM				8 ms	169 ms	2.80 s

## Data Availability

Data are contained within the article.

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
