# Peer review of "Dynamic Golf Swing Analysis Framework Based on Efficient Similarity Assessment"

_sensors, 2025, doi:10.3390/s25227073_

Round 1

Reviewer 1 Report

Comments and Suggestions for Authors

Paper Summary

The paper proposes a simple “dynamic motion similarity” (DMSM) metric for golf swing analysis using 2D pose keypoints. A swing is split into seven phases, and similarity is the area between paired joint trajectories over time. Results show small separation between same-player and different-player pairs, with a short runtime on a standard PC.

Strengths. 
+ The phase-aware design matches how coaches think.
+ The metric is easy to implement, explain, and run on consumer hardware.
+ The figures give an intuitive view of spine-angle trends.

Weaknesses. 
- The dataset is very small (about eight players) and side-view only.
- Gains over a DTW+cosine baseline are tiny and reported without statistics.
- The method description misses key details (segmentation, normalization, joint weighting).
- Some terminology and formatting are inconsistent.

Thank you for submitting to the Sensors journal. The idea is practical and clear, but the current evidence and experimental rigor are not yet sufficient. To improve the manuscript, please justify novelty by comparing with stronger time-series/skeleton baselines, not just DTW+cosine. It is beneficial to include a small user study with coach ratings and report inter-rater agreement, to show the agreement with expert judgement. 

It is better to clarify the dataset and protocol by stating the number of clips per subject, explaining how comparison pairs are formed, and indicating whether evaluation is cross-subject. Also, please strengthen the reporting by adding tables with means and standard deviations across subjects and phases, and ensure all figures have clear axes, units (for example, degrees), and readable legends.

Minor:
Table 3 appears misaligned in the PDF; please fix formatting and present a clear per-stage breakdown.
Abbreviations: “DTW Dynamic Time Wraping” → “Warping.”
Some table/figure formatting appears broken (e.g., Table 3 columns).
Add units for Δ in spine-angle analysis (likely degrees) wherever reported.

Reviewer 2 Report

Comments and Suggestions for Authors

Figure 3 is not clear - Use a single person's data and enlarge the fonts. I understand it's supposed to be representative, so not all numbers need to be clear, but currently, the figure looks cluttered with no meaningful information

Figure 4 is lacking xy-axes
    In the text "Figure 4 compares areas computed with and without interpolation."
    In the label "Figure 4. Comparison of area computed before interpolation."
    Which one is it?

Section 4, it is more accurate to say that the development of the necessary scripts happened in Python 3.10 environment using Anaconda3 as the distribution platform. The way it is written, one might assume that Anaconda is a programming language and not a platform to host and manage Python packages. It's a fairly minor inaccuracy, but the specifics can be important. 
In general, Table 4 should read: Development Environment Python 3.10(Anaconda)
                i3-10100F (CPU), 8GB (RAM) and GTX 1650 (GPU) 
It is also a good idea to include the GPU's VRAM, since pose estimation can be VRAM intensive (I am aware that Blazepose is pretty lightweight though)

The acronym DTW (dynamic time warping) is never explained in the text, it is also never mentioned how it is achieved. This is a pretty standard name and I am going to assume that you used Python's dtwalign library, but it is a good practice to explain all the acronyms and mention the libraries you used.

Figure 7 
It is not clear what it compares (different videos from the same players and then changes to compare different players? Why?)
Why there's an area that is in a red box? (perhaps that can be a separate figure since it compares data from the same player) 
Most of the bar graphs are not addressed in the text? What argument is being supported that is not addressed by the area in the red box?

In general Figure 7 appears to take up too much space without providing an equal amount of information. If I understand  your argument correctly, the comparison of different videos for each player is sufficient to show the shortcomings of the cosine similarity

Figure 8 is blurry, also consider using different line styles in addition to colour ( e.g red + ---, blue + ****) to help with clarity.

Figure 10 one would expect that the proposed Dynamic Motion Similarity Measurement would actually be closer to 1 for different videos of the same person. At the same time, how it is possible for something to achieve similarity >1? Please explain how the proposed DMSM is supposed to be interpreted. To make my point perfectly clear, from Eq. 1, it is obvious that the closer to 0, the more similar two trajectories are, but Similarity = 0 from figure 10, conveys the wrong meaning to the reader. Figure 11 that uses "difference" for the y-axis, is easier to interpret.

Consider converting your bar graphs to Tables to make the results clearer.

Reviewer 3 Report

Comments and Suggestions for Authors

The manuscript presents a Dynamic Motion Similarity Measurement (DMSM) framework designed for golf swing analysis using video-based pose estimation.

The DMSM framework is conceptually sound, but its novelty relative to existing DTW-based or trajectory-based similarity measures is not fully established. The authors should clearly articulate how DMSM differs mathematically and functionally from conventional temporal alignment or motion trajectory metrics.

The paper’s contribution is primarily methodological integration rather than algorithmic innovation. To justify publication in Sensors, the authors should emphasize the practical advantages or measurable improvements  that make DMSM superior to previous techniques.

The dataset used for validation is too limited to draw general conclusions. Please provide the total number of pairwise comparisons and statistical analysis. Include cross-validation or bootstrapping to ensure reproducibility.

The reported improvements are extremely small and may not be statistically significant. The authors should conduct a significance test to confirm that the observed gains are meaningful.

There is no ground truth evaluation linking DMSM scores to expert or coach-assigned similarity ratings. Without such validation, claims about “alignment with expert perception” remain speculative.

Equation (1) defines a simple integral of absolute differences, which is conceptually similar to the L1 distance between trajectories. However, there is no weighting or normalization to account for different motion amplitudes or coordinate scales. Please clarify whether normalization is applied. Explain how 2D projection distortion affects DMSM’s accuracy, especially since only side-view data are used.

The segmentation into seven phases is central to the paper but lacks automated boundary detection details. What criteria define transitions between phases? Are these rules robust across players and camera setups?

The computational efficiency is only marginally slower than DTW. However, the hardware configuration and model inference times are outdated benchmarks; it would be useful to report FLOP complexity or expected inference time on CPU-only devices.

The paper compares only against “DTW + cosine similarity,” which is insufficient. For a fair assessment, include additional baselines, such as:
Euclidean trajectory distance,
phase-wise correlation metrics,
PSIM (Pose Similarity Metric),
or other motion energy–based similarity measures.
Without these, it is difficult to evaluate DMSM’s true improvement beyond incremental performance gains.

The discussion section is mostly descriptive and lacks quantitative interpretation of biomechanical implications. Provide explicit examples linking DMSM features to known golf swing mechanics. Explain how coaches could use DMSM feedback in practical training scenarios.

The claim that DMSM “aligns with expert perception” should be supported by qualitative validation.

Reviewer 4 Report

Comments and Suggestions for Authors

The topic of this article is interesting. The article contains all the necessary components required for a scientific paper. However, some aspects require further clarification:

  1. Figure 2. The information in this figure is essential and important for this research. Would it be possible to split the figure to better see and understand the body position in each stage? This would make it clearer to see the differences from one position to another.
  2. Fig.2..... Is it from a professional or amateur?
  3. See all the graphs. In my view, all the graphs are very important.
    My suggestion is: would it be possible to reconsider the representation?
    Instead of having the records for all participants in the same graph,
    perhaps split them: amateurs and professionals, or between two levels of professionals,
    or by gender.
    It will be very interesting for readers to notice the differences,
    and for the authors as well to extend the conclusion section.
  4. What are the limitations of this study? What further research can be conducted on the proposed method?
  5. I suggest extending the Conclusion section.

Round 2

Reviewer 1 Report

Comments and Suggestions for Authors

The authors have strengthened the paper in several ways. They expand preprocessing/normalization and add additional baselines. The figures now annotate units for spine angle (degrees), and the text notes mean±SD. Minor issues have mostly been cleaned up. However, several figures quality must be improved before publishing.

Reviewer 3 Report

Comments and Suggestions for Authors

The study proposes a dynamic similarity metric that is technically sound but only moderately novel. The authors should clearly distinguish DMSM from existing trajectory-based and DTW-aligned methods and highlight the theoretical advantage of combining biomechanical phase-awareness with trajectory integration.

The dataset is too small for strong statistical generalization, and details on data distribution, cross-validation, and sampling are missing. The authors should provide complete information on player counts, evaluation reliability, and include effect sizes and confidence intervals for statistical tests.

The comparison with DTW plus cosine similarity lacks depth and completeness. Quantitative results for additional baselines such as PSIM and Euclidean distance should be reported, and all methods should be summarized in a unified performance table with accuracy and efficiency metrics.

The analysis is limited to side-view data, which restricts generalizability. The authors should discuss robustness to camera angle, frame rate, and pose estimation errors and report performance variance across multiple trials to demonstrate stability.

The reported runtime suggests the method is not yet real-time, and the GPU performance inconsistency requires clarification. The authors should explain this issue and discuss possible code optimizations to support real-time deployment.

Figures are informative but lack quantitative clarity, proper labeling, and sufficient visual contrast. The discussion should more directly connect observed numerical trends to underlying biomechanical insights to enhance interpretability.
